# Applying the Techniques of Materials Science towards an Understanding of the Process of Canine Intervertebral Disc Degeneration

**DOI:** 10.3390/ani14182665

**Published:** 2024-09-13

**Authors:** Viviana Rojas, Ravin Jugdaohsingh, Andrew Rayment, Andrew Brown, Joseph Fenn, James Crowley, Vedran Lovric, Jonathan Powell, Paul Freeman

**Affiliations:** 1Department of Veterinary Medicine, University of Cambridge, Cambridge CB3 0ES, UK; vr353@cam.ac.uk (V.R.); rj328@cam.ac.uk (R.J.); jjp37@cam.ac.uk (J.P.); 2Department of Materials Science and Metallurgy, University of Cambridge, Cambridge CB3 0ES, UK; awr22@cam.ac.uk; 3School of Chemical and Process Engineering, University of Leeds, Leeds LS2 9JT, UK; 4Clinical Sciences & Services, Royal Veterinary College, London NW1 0TU, UK; jfenn@rvc.ac.uk; 5Small Animal Specialist Hospital, Alexandria, Sydney, NSW 2015, Australia; james.crowley@hotmail.com; 6Surgical and Orthopaedic Research Laboratories, Prince of Wales Clinical School, Faculty of Medicine, University of New South Wales, Sydney NSW 2052, Australia

**Keywords:** intervertebral disk, Hansen, disc disease, extrusion, spinal cord, calcification, atomic force microscopy

## Abstract

**Simple Summary:**

Intervertebral disc disease is a very common problem in dogs. The disease causes pain and paralysis and leads to the death of many dogs annually. The disease is intimately associated with disc degeneration and calcification, but the precise mechanisms by which these pathological processes are linked remain a mystery. Most studies to date have focused on imaging changes and histopathology of affected discs, but here we take a novel approach exploring the use of a number of different techniques associated with mineralogy and materials science, including atomic force microscopy and scanning and electron microscopy, and assessing the information they are able to provide. Our results have allowed us to generate some testable hypotheses which could ultimately help to unravel the way in which disc degeneration and disease are linked, and lead to novel methods of prevention and treatment.

**Abstract:**

Intervertebral disc degeneration in dogs occurs in an accelerated way and involves calcification, which is associated with disc herniation or extrusion. The degenerative process is complex and involves the transformation of collagen fibres, loss of proteoglycans and notochord cells and a reduction in water content; however, how these processes are linked to future disc extrusion remains unknown. We have employed techniques including Fourier Transform Infra-red Spectroscopy (FTIR), Scanning Electron Microscopy (SEM), Transmission Electron Microscopy (TEM), Uniaxial Compression Loading and Atomic Force Microscopy (AFM) in an attempt to gain a greater understanding of the degenerative process and its consequences on the physical properties of the disc. FTIR verified by TEM demonstrated that calcium phosphate exists in an amorphous state within the disc and that the formation of crystalline particles of hydroxyapatite occurs prior to disc extrusion. AFM identified crystalline agglomerates consistent with hydroxyapatite as well as individual collagen fibres. SEM enabled the identification of regions rich in calcium, phosphorous and oxygen and allowed the visualization of the topographical landscape of the disc. Compression testing generated stress/strain curves which will facilitate investigation into disc stiffness. Ongoing work is aimed at identifying potential areas of intervention in the degenerative process as well as further characterizing the role of calcification in disc extrusion.

## 1. Introduction

Intervertebral disc disease (IVDD) is the most common cause of spinal cord injury in dogs [1]. The disease is seen most commonly in short-legged breeds known as chondrodystrophic, such as the dachshund and French Bulldog [1], but it can occur in any breed with potentially catastrophic consequences. The commonest form of the disease, and that which occurs most frequently in the chondrodystrophic breeds, is disc extrusion, where calcified material from the inner part of the disc or nucleus pulposus breaks through the outer shell or annulus fibrosus to impact the spinal cord lying dorsal to the disc [2]. The treatment of disc extrusion in dogs can involve decompressive surgery, where the vertebral canal is opened and the extruded calcified material is removed from around the spinal cord [3]. At the same time, surgeons often carry out a prophylactic procedure known as fenestration, which involves an attempt to evacuate material from the nucleus pulposus in order to prevent future extrusion [4].

A genetic factor known as the FGF4 retrogene on chromosome 12 has been demonstrated to be associated with this form of IVDD as well as chondrodystrophy in dogs [5], but breeding away from the disease has thus far not been possible due to the apparently ubiquitous distribution of this retrogene in susceptible breeds [6].

The intervertebral disc consists of three main components. The outer fibrous annulus fibrosus (AF) has a lamellar structure made up of layers of fibrous tissue rich in collagen types I and II. The inner gelatinous nucleus pulposus (NP) contains mainly notochordal cells in an extracellular matrix (ECM) rich in high molecular weight proteoglycans and collagen type 2 fibrils [7]. This arrangement creates an osmotic pressure, which ensures the NP retains a high water content, conferring a gelatinous nature important for the shock-absorbing function of the disc. Chondrocytes are also present within the NP, arranged mainly in pairs or strings and surrounded by a pericellular matrix, which has been shown to be softer than the ECM and is thought to be protective of the chondrocytes when the disc is subjected to normal weight-bearing forces [8]. The disc is then bounded by cartilaginous endplates, which bind it firmly between adjacent vertebrae and are the source of nutrition for the disc since the AF and NP are avascular structures.

IVDD in dogs is associated with disc degeneration and especially the calcification of the NP, which occurs commonly and early in life in chondrodystrophic breeds. Disc degeneration is a complex and multi-faceted process, which is still poorly understood and may hold the key to more effective prophylaxis and treatment protocols. It is known that the NP loses notochordal cells, with clusters of chondrocytes becoming the predominant cell type. Collagen type I fibres become more common and type II less common in the NP as degeneration progresses, and there is a loss of proteoglycans, leading to a reduction in osmotic pressure and, thus, water content [7]. Calcification may be found within the NP in dogs as young as 6 months, and a higher number of calcified discs identified on radiographs in dachshunds has been associated with an increased risk of disc extrusion later in life [9].

We have investigated the calcium phosphate signature in the NP of both extruded and non-extruded canine intervertebral discs using Fourier transform infra-red spectroscopy (FTIR) and transmission electron microscopy (TEM). FTIR allows the determination of underlying bands associated with specific atomic bonds in the spectra of biological materials. In the current work, we focused on the V1, V3 phosphate contour (900–1200 cm^−1^) as it is readily accessible and has previously been shown to be a useful method of assessing the presence of crystalline hydroxyapatite [10].

In addition, we have taken a novel approach by employing techniques from materials science in an attempt to further unravel the degenerative process. Materials science is a discipline which strives to link together the structure of materials, often at the nano- and micro-scale, with their properties and the processes by which they are formed. Here, we focus particularly on structure–property links in the nucleus pulposus. Atomic force microscopy (AFM) has been used for a number of years to provide nanoscale topographical and mechanical details of many biological tissues. An early study used AFM to analyze some of the characteristics of rabbit AF [11], and the technique was later used as part of the analysis of a rabbit degenerative disc model [12]. More recently, AFM has been used to characterize some details of the extracellular matrix (ECM) of mouse model intervertebral disc degeneration [13], in particular collagen D-spacing, and was also used to measure changes and variations in stiffness of the ECM and pericellular matrix. We report the use of this technique for the first time to investigate the properties of the degenerate canine NP, including the finding of regions containing sharp-edged crystalline structures and areas where collagen fibres are present.

Scanning Electron Microscopy (SEM) is a further technique able to give topographical and biochemical information and has been used to examine the rat intervertebral disc [14]. A number of studies have since reported the use of SEM in the analysis of human disc samples removed at surgery [15,16], as well as specifically looking at details of the bone and cartilage endplates [17]. Here, we report the first use of SEM for analysis of the canine NP, specifically using the technique to identify regions rich in calcium phosphate.

Uniaxial load compression (with a microindenter) has been used mainly to test the stiffness and micromechanical properties of whole disc specimens from post-mortem small animal models. One such study reported increasing stiffness of the NP after loading in a rat IVDD model [18]. We have used compression testing to generate stress/strain curves that allow comparison of stiffness between different samples.

The aim of this ongoing work is to elucidate aspects of the degenerative process underlying canine IVDD. We report the findings of a number of novel techniques we have used in the analysis of various samples of canine nucleus pulposus in an attempt to characterize the changes occurring within the disc prior to extrusion.

The current study was approved by the Ethics and Welfare Committee of the Department of Veterinary Medicine, University of Cambridge CR442.

## 2. Materials and Methods

### 2.1. Sample Collection

#### 2.1.1. Population

Dogs that were presented to the neurology services at three different veterinary hospitals: Queen’s Veterinary School Hospital, University of Cambridge; Small Animal Specialist Hospital, North Ryde, NSW; and Queen Mother Hospital for Animals, Royal Veterinary College, London, and had decompressive spinal surgery because of intervertebral disc extrusion.

#### 2.1.2. Inclusion Criteria

Dogs of any breed, size and age with a complete medical history who had a diagnosis of intervertebral disc extrusion. The dogs were examined by a board-certified neurologist or surgeon or a neurology resident working under the supervision of a board-certified neurologist, had the diagnosis confirmed by MRI and/or CT, and had surgical decompression as treatment and to confirm the presence of an extrusion.

#### 2.1.3. Sample Collection

Extruded IVD materials were collected during decompressive surgery (hemilaminectomy or mini hemilaminectomy for thoracolumbar extrusions and ventral slot for cervical) directly from the exposed vertebral canal at the site of extrusion. These samples were labelled ‘extruded material’.

Non-extruded IVD materials (nucleus pulposus) were collected through the prophylactic fenestration of affected and adjacent discs during the decompressive surgery. This procedure was performed with a scalpel blade and curette. Samples were collected using different surgical instruments for each type of sample in order to avoid cross-contamination. The samples extracted from adjacent non-extruded discs by this process were labelled ‘non-extruded NP’, and the samples extracted from affected (extruded) discs by the same process were labelled ‘extruded NP’).

Following surgical collection, most samples were transferred to an empty sterile plastic universal container before being transferred to a −20 °C freezer. When freezing was not possible immediately after surgery, the disc materials were initially stored at +4 °C and then transferred to the freezer within a few days.

The results reported represent an investigation of a range of different samples without an attempt to cross-correlate results, with the exception of FTIR and TEM. A distinction between extruded and non-extruded samples is also only reported for the FTIR analysis in the current study.

Clinical data, including age, sex, breed, and neurological status, as well as imaging data, including MRI or CT appearance of affected discs, was recorded, but only summary age and breed characteristics are reported for context.

### 2.2. Freeze-Drying Process

After a minimum of 24 h in the freezer (−20 °C), all the samples underwent freeze drying using the Christ^®^ Alpha 1–2 LDplus Freeze Dryer (Christ, Osterode am Harz, Germany). The freeze dryer was warmed up for at least 15 min prior to the start of the main drying phase, and at the same time, the ice condenser pre-cooled (“cool-down”) to bind the released water vapour. The lids of the sample containers were removed and the disc samples in the containers were placed on shelves of the freeze dryer. The main drying phase, where the sublimation occurs, was completed in 5 h, followed by the second phase (final drying). The final drying process, carried out in order to remove any residual moisture, occurred over 15 h according to the manufacturer’s instructions.

### 2.3. Fourier-Transform Infra-Red Spectroscopy (FTIR)

Most samples were analyzed using FTIR following freeze-drying in order to minimize water signal effects. FTIR analysis was performed using the Shimadzu^®^ IRPrestige21 (Shimadzu, Tokyo, Japan) set to a scan speed of 2.8 mm/s and a resolution of 2 cm, covering wavenumbers 400 to 4000 cm^−1^. Before each analysis, the sample holder was cleaned with 100% ethanol and a background scan was performed. Samples were then scanned between one and three times, depending on size. No grinding of samples was performed.

#### 2.3.1. Spectral Pre-Processing

Signal processing was performed on all FTIR spectra of IVD samples using MATLAB^®^ version R2024B (The MathWorks, Inc., Los Angeles, CA, USA). This encompasses smoothing by application of a Savitzky–Golay filter and derivative spectroscopy to isolate specific vibrations of chemical interest. Both extruded (sample) and non-extruded (control) spectra were then observed for the visual appearance of their PO_4_^3−^ V3 absorption peak and the presence or absence of the PO_4_^3−^ V1 absorption peak.

#### 2.3.2. Interpretation of FTIR Spectra

The smoothed spectrum of each sample was visually evaluated, and based on the presence or absence of phosphate V1 and V3 vibration peaks in the region 900–1200 cm^−1^ (Figure 1), each sample was qualitatively classified as containing or not containing phosphate and containing or not containing significant crystalline hydroxyapatite (Appendix A).

### 2.4. Transmission Electron Microscopy (TEM)

Five representative samples previously analysed using FTIR were submitted for TEM analysis in order to provide verification of FTIR analysis results. Samples were prepared by suspending a small piece of the freeze-dried material in ethanol, grinding in a pestle and mortar and then drop-casting finely suspended pieces direct onto holey carbon support films (EM Resolutions Ltd., Newcastle, UK) and allowing them to air dry before transfer into the TEM.

Analysis was carried out using FEI Titan^3^ Themis 300 (Thermofisher Scientific, Waltham, MA, USA): X-FEG 300 kV S/TEM with S-TWIN objective lens, monochromator (energy spread approx. 0.3 eV), multiple HAADF/ADF/BF STEM detectors, FEI Super-X 4-detector EDS system, Gatan Quantum ER energy filter for EELS and EFTEM and Gatan OneView 4K CMOS digital camera (Gatan Inc., Pleasanton, CA, USA).

### 2.5. Atomic Force Microscopy (AFM)

Samples for AFM analysis were smeared onto a microscope slide immediately after collection during the surgical procedure and not subjected to the freezing process above. The topographical assessment was made using AFM in peak force tapping mode using a Bruker Dimension Icon Pro^®^ microscope (Bruker, Karlsruhe, Germany). Different samples were evaluated using two types of silicon AFM probes: RTESPA-525 (spring constant 200 N/m) and SCANASYST—AIR (spring constant 0.4 N/m) from Bruker. Images were processed using the Bruker NanoScope Analysis v1.40r1Software^®^. Once the sample was located under the microscope, multiple random regions of each sample were scanned in order to recognise the physical features of the surface and identify structures described previously in collagen-based tissues (e.g., collagen fibres and “D-spacing” [13]) and possible structures compatible with crystal habits of minerals, e.g., hydroxyapatite crystal habit [19].

### 2.6. Scanning Electron Microscopy (SEM)

A number of freeze-dried samples from extruded and non-extruded discs were directly mounted on SEM specimen stubs of 12.5 mm diameter and sputter coated with gold. The analysis was performed using the Hitachi^®^ TM 4000 plus (Hitachi, Tokyo, Japan) tabletop microscope, and image maps of the surface of each sample were obtained at a range of microscales. Energy Dispersive X-ray Spectroscopy (EDS) layered images from those regions were obtained, and exploration of the sample surface was performed by looking for areas of interest with the presence of significant agglomerations of calcium, phosphate and oxygen. The mineral proportion of each sample was represented in a map-summed spectrum for each area scanned.

### 2.7. Uniaxial Load Compression (ULC)

Fresh or fresh-frozen samples were subjected to uniaxial strain/stress compression testing using the 1ST Tinius Olsen Universal Testing Machine, with a calibrated 25 N load cell (class 0.5) and controlled by the computer’s Horizon data analysis software. Samples of extruded and non-extruded discs were tested. The samples were transported in a polystyrene container in order to maintain them at a low temperature of around 4C following removal from refrigeration. The cross-sectional areas of the samples evaluated were calculated after placing the samples on a 1mm scale graph paper. Samples were then placed in a compression grip without extra moisture, and the tests were conducted at a displacement rate of 2 mm/min until the specimen was compressed to a nominal 80%. The thickness was calculated during the compression loading by recording the value at the time when the first force value was documented.

Stress/strain curves were obtained from each measurement. All curves were plotted on the same graph allowing comparison of slopes as an indicator of stiffness.

## 3. Results

### 3.1. Fourier Transform Infra-Red Spectroscopy (FTIR)

A total of 55 samples of extruded and non-extruded nucleus pulposus (NP) were tested taken from 34 individual dogs, including 29 samples of extruded material, 9 samples of NP retrieved from extruded discs by fenestration, and 17 samples of NP taken from non-extruded discs also by fenestration. Breeds included eight Miniature Dachshunds, eight French Bulldogs, five Cocker Spaniels, two Cavalier King Charles, two Pekingese, one Jack Russell Terrier, one Chinese Crested, one Lhasa Apso, one Miniature Schnauzer, one Springer Spaniel, one Beagle, one Irish Terrier, one Maltese and one Terrier X. The age range was 3–13 years. The presence or absence of significant crystalline hydroxyapatite (HA) is recorded in Table 1 below based on the presence or absence of a PO_4_^3−^ V1 vibration peak in the 955–965 cm^−1^ region on the FTIR smoothed spectra (see Figure 1). In total, 23/55 (41%) samples were classified as containing significant crystalline HA and 32/55 (59%) as not containing significant crystalline HA [19].

In total, 18/29 (62%) of the extruded samples were positive for significant HA versus 11/29 (38%) negative. Of the non-extruded NP samples, 2/17 (12%) contained significant HA and 15/17 (88%) were negative (Table 1).

### 3.2. Transmission Electron Microscopy (TEM)

Five samples of extruded and non-extruded nucleus pulposus previously analysed with FTIR were examined. Gel-like deposits were found on grids for all samples examined. For three of the five samples, the gel deposits contained particles of nanoscale, needle-like agglomerates (red arrows in Figure 2A). Electron diffraction from the gel showed rings of diffraction spots, typical of a polycrystalline material, and spacing consistent with those expected for hydroxyapatite, with the two most intense rings at ~3.6 nm^−1^ and ~2.9 nm^−1^ (corresponding to real-space distances of ~2.8 and 3.5, Angstrom, respectively) always visible (Figure 2B: blue arrows) [20]. The reference pattern used for this characterisation was electron diffraction measured at 200 kV from a hydroxyapatite standard powder [21]. Energy-dispersive X-ray spectroscopy (EDS) showed high Ca, P and O signals. Of these three samples, two were characterized as positive for significant HA with FTIR and one as negative.

For the remaining two samples, the gel deposits contained relatively fewer particles (Figure 3A), and electron diffraction confirmed the gel was predominately amorphous (Figure 3B). EDS further suggested the gels were Si- and O-rich, but some Ca and P were also detected, consistent with the presence of some needles in the gel. Of these two samples, one was characterized as HA-rich with FTIR and one was not.

### 3.3. Atomic Force Microscopy (AFM)

Topographical characterization of directly smeared samples of extruded nucleus pulposus revealed structures compatible with collagen fibres based on previous reports [22] (Figure 4). Due to the heterogeneity within the nucleus pulposus, it was necessary to scan multiple areas in order to identify these structures. Collagen fibrils are several micrometres in length and can be identified by their characteristic, periodically banded pattern, also known as D-banding [23].

Using AFM peak force mapping mode, topographical visualisation of one sample at the nanoscale level revealed areas with a topography compatible with aggregates of particles in the shape of hexagonal prisms with sharp corners and edges (Figure 5), which are consistent with aggregations of crystals whose habits are hexagonal, such as hydroxyapatite [24].

### 3.4. Scanning Electron Microscopy (SEM)

A total of eight samples were scanned. All the samples were extruded materials.

Map Sum Spectrum from the energy dispersive X-ray spectroscopy (EDS) analysis was used to characterize the proportions of the target elements. Figure 6A shows the sum of spectral data recorded across the map from a sample, which, prior to FTIR analysis, was suggestive of crystalline HA (see also Figure 7A,B) versus Figure 6B, which is from a sample that did not reveal crystalline HA on FTIR (see also Figure 7C,D). It is clear that both samples contain significant minerals and indeed appear very calcium-rich compared to the accepted standard calcium:phosphorous ratio for hydroxyapatite of c.1.7. However, it has been shown that factors such as particle size and electron beam acceleration can affect this ratio [25]. Furthermore, these results are from map sum spectra of the entire scanned image, and we aim to refine this in future by utilizing multiple single-point spectral analyses. If these results are repeatable, it would indicate an extremely calcium-rich NP.

Secondary electron images were obtained after scanning the samples. EDS-layered images were used to identify and characterize the distribution of the Ca, P and O (see Figure 7), whose presence in the sample was previously confirmed using FTIR.

### 3.5. Uniaxial Load Compression Testing

Stress/strain curves were obtained from 23 different samples, most of them extruded material plus a small number of non-extruded samples from fenestrations of adjacent discs. Cross-sectional areas of samples tested ranged from 13 mm^2^ to 167 mm^2^, with an average area of 41.4 mm^2^. Figure 8 illustrates the stress/strain curves of all the samples tested. It is clear that a majority of samples tested behave in a very similar manner in response to compression, with the exception of a small number of samples, which appear to be significantly stiffer.

## 4. Discussion

We have shown, with Fourier transform infra-red spectroscopy, that most samples of extruded intervertebral disc material contain significant crystalline hydroxyapatite (HA), whereas this is the case for only a small proportion of non-extruded samples. We have further shown, using a combination of scanning and transmission electron microscopy (TEM), that most of the non-extruded samples do still contain calcium phosphate but that this is present in an amorphous form, often with needle-like crystals occurring within the amorphous matrix. We therefore hypothesize that transformation from the less stable amorphous calcium phosphate (ACP) to the more stable HA is likely occurring and possibly necessary prior to disc extrusion. Failure to confirm the presence of crystalline HA in all extruded samples is expected due to a combination of the qualitative nature of FTIR analysis and the degree of subjectivity necessary in the interpretation of smoothed spectra [10]. In addition, the heterogeneity of the samples collected and the reliance on clinically obtained material with the potential to contain contaminants such as blood and other soft tissues are also likely to affect these results. The presence of some crystalline HA in a small proportion of non-extruded samples is also consistent with this hypothesis if there is a gradual transformation from ACP into crystalline HA prior to disc extrusion [19,20].

The process of calcification is poorly understood. We have previously shown that in diseased discs, the extruded NP contains mainly crystalline hydroxyapatite, whereas many non-extruded but calcified discs contain an amorphous phase of calcium phosphate [26]. This leads us to speculate that a phase shift in the calcium phosphate found in the degenerate NP may predispose the disc to herniate or extrude. Why this should be the case is not known, but it might reasonably be associated with both increased stiffness and a loss of shock-absorbing properties of the disc, as well as a physical degradation of AF fibres by the needle-like hydroxyapatite agglomerates. In addition, the AF itself undergoes degenerative processes predisposing to cracks and splits developing associated with changes in collagen fibre type and orientation [7,14,16].

We have further used TEM to verify the interpretation of FTIR curves and demonstrated that all extruded samples examined with this technique contain a mixture of gel-like amorphous material as well as some needle-like crystalline HA. The proportions of these two components vary significantly, with some containing predominantly amorphous material alone and others having a high concentration of crystalline material, implying a gradual shift from ACP to crystalline HA. It is, of course, possible that due to the extremely small regions of analysis made with TEM, the variation in results represents differences in different regions of the same sample; this seems unlikely given the significantly higher proportion of samples showing the presence of crystalline material in extruded samples versus non-extruded found with FTIR analysis. We hypothesize that the formation of sufficient amounts of crystalline HA with large enough particle sizes within the NP makes the disc less able to withstand the forces applied to it during everyday movements [18] and that this contributes to the reason why certain discs are prone to extrusion. We have used uniaxial load compression testing to generate stress/strain curves and demonstrated that this may be a useful technique with which to examine this possibility.

To the knowledge of the authors, this is the first report of the use of atomic force microscopy (AFM) to analyze canine intervertebral disc nucleus pulposus [11,12,13]. In our investigation, a variety of protocols were tried in order to prepare the most suitable specimen for AFM testing, including agar/gelatine phantom models and flash-frozen OCD cryocuts, but none of these approaches has so far provided superior results to the original basic approach of smearing the material directly on a microscope slide. Significant challenges were encountered, including selecting representative regions of the different samples to scan and produce preparations, which allowed reliable results with the use of peak force tapping mode, and many attempts were made with significant difficulties in obtaining consistent topographical or numerical results. We have, however, shown that it is possible, although not easy, to prepare samples in a way that is suitable for the use of AFM analysis and that this technique may be able to identify crystal agglomerates as well as other components of the degenerate NP which may play a role in extrusion such as collagen fibres. Interestingly, neither AFM nor SEM were able to demonstrate the presence of the needle-like crystalline particles, as evidenced by TEM. This is probably due to the fact that both are surface analysis techniques, unlike TEM, and it is likely these needles are buried within the matrix. In TEM, electrons are able to pass through the sample to reveal the needles [27]. Furthermore, distinguishing these small needles against a background of large roughness may be very difficult. AFM was, however, able to identify larger-surface crystalline agglomerates, which are potentially too large to be imaged with TEM and likely not electron-transparent. It is possible that the needle-like crystals seen with TEM are precursors of the larger agglomerates seen with AFM and that the process of pre-extrusion calcification may involve the initial formation of small needle-like hydroxyapatite crystals, which mature into larger agglomerates. The heterogeneity and size of samples for analysis means AFM will remain a challenging technique but nevertheless an area worthy of further investigation, with closer analysis of collagen fibrils potentially revealing further insights into the degenerative process.

Scanning Electron Microscopy (SEM) is a versatile tool for mineralogical analysis, providing valuable information about the surface morphology, crystallography and elemental composition of minerals [15,27]. SEM analysis of these samples was valuable in confirming the presence of the elements of Ca, O and P and showing their distribution, as well as giving some information regarding the surface topography of the samples on a larger scale than AFM. The ratio of calcium to phosphorous was higher than expected in some samples, and this requires further investigation, as discussed above. The use of SEM further enabled the confirmation of the presence of calcium phosphate in samples that did not contain overtly crystalline hydroxyapatite. We feel that this technique shows promise in the analysis of clinical samples of nucleus pulposus since the samples may not require such meticulous preparation as AFM and are less subject to the problems of irregular surface topography and heterogeneity. The technique may further be useful in identifying regions suitable for more detailed examination using AFM.

Finally, through the use of uniaxial load compression testing, we have been able to generate stress/strain curves which appear to show that most samples of NP behave in a similar manner in response to compression testing [18]. This may suggest that a change in stiffness does not play a significant role in disc extrusion, however, testing of a larger number of samples with a known mineral analysis will be required to further investigate this possibility.

### Limitations of This Study Include

1. The heterogeneity inherent in sample acquisition.

2. Difficulties encountered in suitable sample preparation, particularly for AFM.

3. Lack of previous work with which to compare our findings.

## 5. Conclusions

We have confirmed the presence of crystalline hydroxyapatite in samples of extruded material following canine intervertebral disc extrusion, and also that many non-extruded discs contain an amorphous version of calcium phosphate. We have been able to image the surface topography of nucleus pulposus samples with AFM and SEM, as well as using SEM to confirm the presence of significant minerals in these samples. Through TEM, we have further confirmed that the calcified nucleus pulposus contains needle-like particles of crystalline hydroxyapatite within an amorphous material, all of which is consistent with our hypothesis of a shift from amorphous calcium phosphate to crystalline hydroxyapatite prior to disc extrusion. The use of uniaxial compression testing has further enabled the quantification of the stiffness of the material, which may be relevant to the process of disc extrusion. Further work is necessary to establish the precise role of calcification in intervertebral disc extrusion in dogs, with the aim of intervening to reduce or prevent the disease from occurring.

## Figures and Tables

**Figure 1 animals-14-02665-f001:**
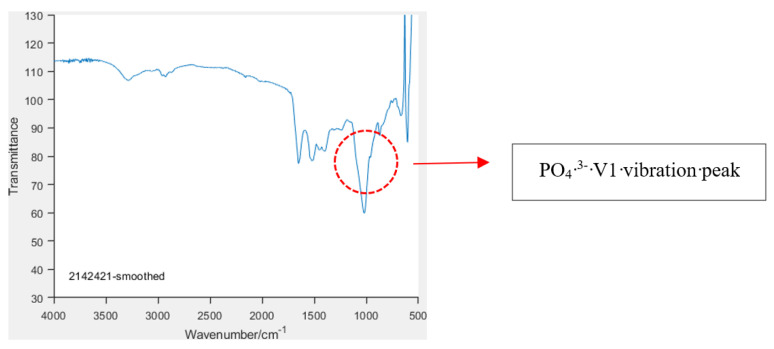
Graph of typical smoothed spectra from an extruded intervertebral disc material, showing the PO_4_^3−^ V1 vibration peak (encircled region) characteristic of hydroxyapatite mineral content.

**Figure 2 animals-14-02665-f002:**
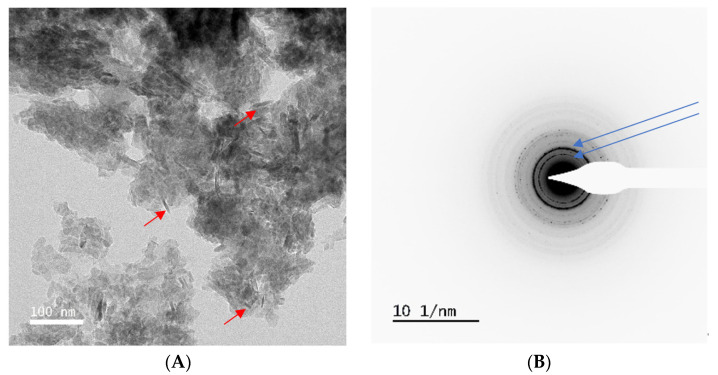
(**A**) TEM image of gel with many needle particles. (**B**) Polycrystalline electron diffraction (sharp rings).

**Figure 3 animals-14-02665-f003:**
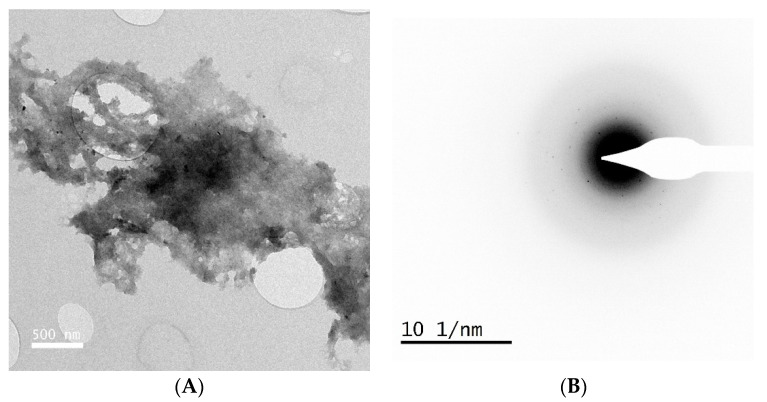
(**A**) TEM image of gel with very few needle particles. (**B**) Amorphous electron diffraction (no sharp rings).

**Figure 4 animals-14-02665-f004:**
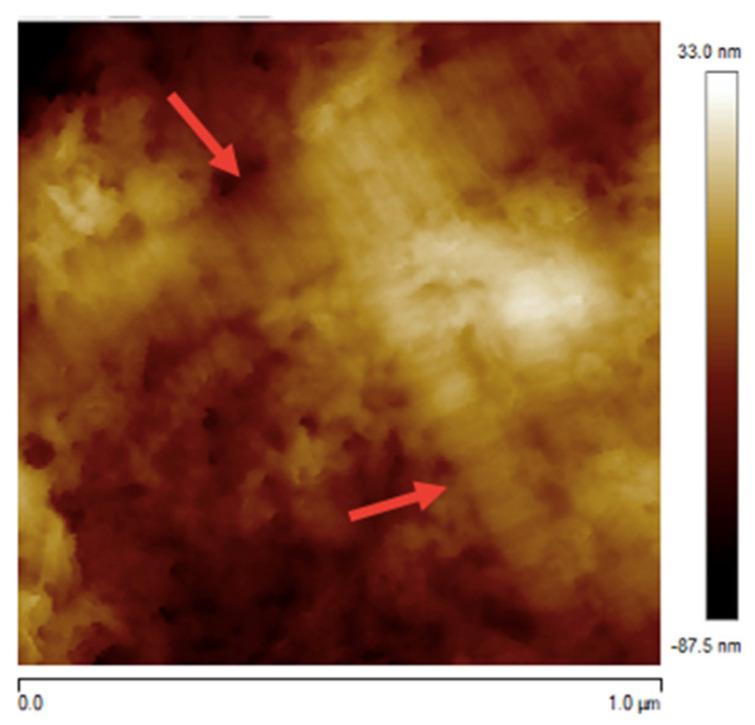
Structures compatible with collagen fibres (red arrows) in extruded material at a scan size of 1000 nanometres.

**Figure 5 animals-14-02665-f005:**
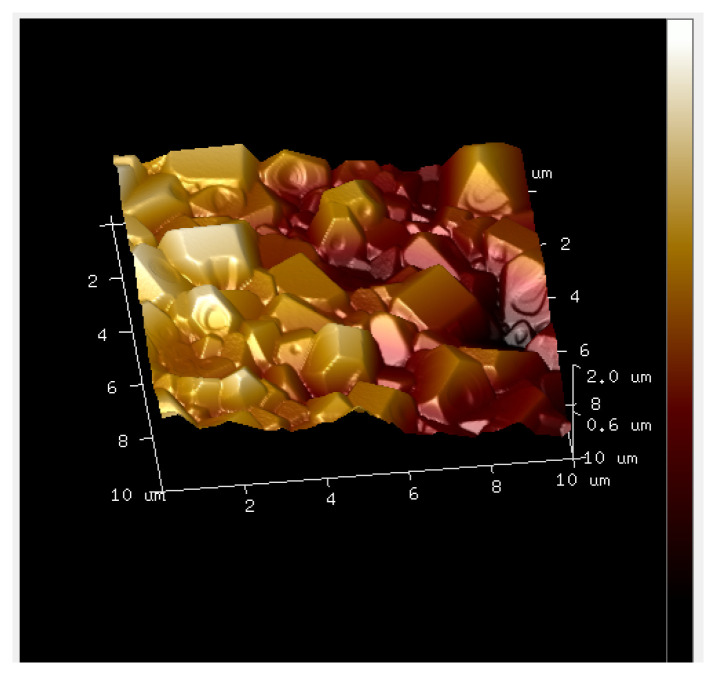
“Hexagonal-like” particles aggregated. Topographical approach at a nanoscale of 10 µm.

**Figure 6 animals-14-02665-f006:**
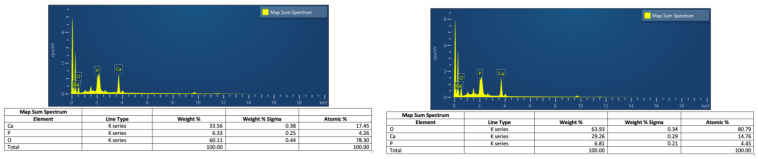
EDS Map Sum Spectrum from samples containing Crystalline HA Calcium (**A**) and without Crystalline HA Calcium (**B**) based on FTIR spectra (see Figure 7).

**Figure 7 animals-14-02665-f007:**
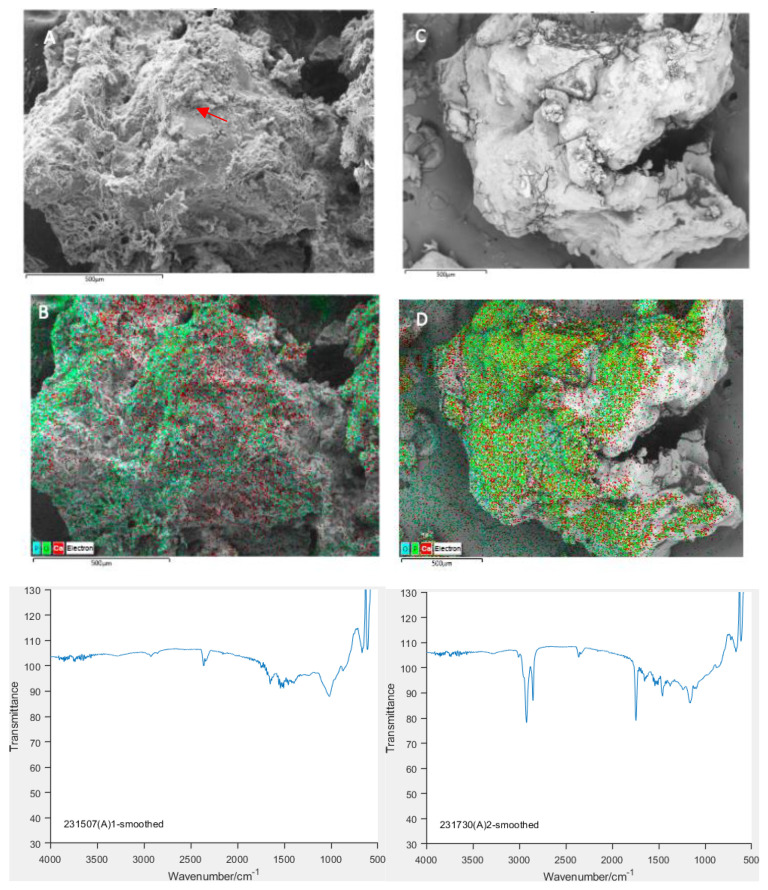
The images (**A**,**C**) represent the electron images from the 2 samples represented in Figure 6. The images (**B**,**D**) are the respective EDS-layered images showing the distribution of the Ca, P and O in the surface. The smoothed FTIR graph for each sample is presented below the SEM images; the V1 vibration peak consistent with presence of HA can be clearly identified only in the first of the 2 spectra (red arrow).

**Figure 8 animals-14-02665-f008:**
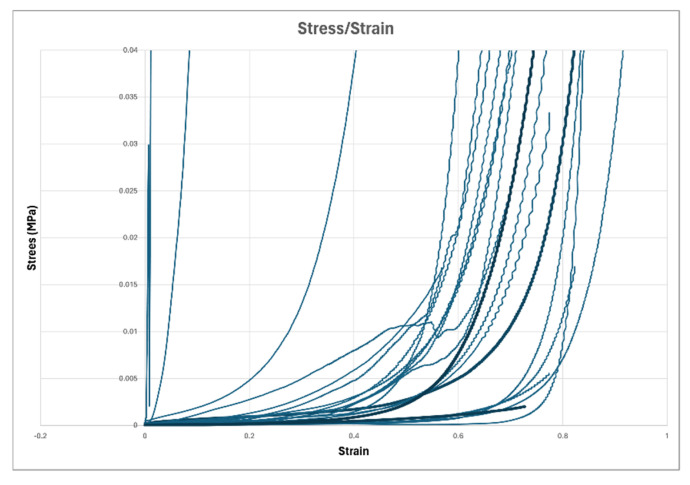
Stress/strain curves from uniaxial load compression testing.

**Table 1 animals-14-02665-t001:** Results of visual inspection of FTIR-smoothed spectra for presence/absence of V1 peak.

Type of Sample	Crystalline Hydroxyapatite	
	Yes	No
Extruded material (29)	18 (62%)	11 (38%)
Extruded disc nucleus pulposus (9)	3 (33%)	6 (67%)
Non-extruded disc nucleus pulposus (17)	2 (12%)	15 (88%)

## Data Availability

All original data forms part of an ongoing PhD study and is available on request from the corresponding author.

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
