# Peer review of "Applying the Techniques of Materials Science towards an Understanding of the Process of Canine Intervertebral Disc Degeneration"

_animals, 2024, doi:10.3390/ani14182665_

Round 1

Reviewer 1 Report

Comments and Suggestions for Authors

This study primarily established the presence of calcium phosphate, either in the amorphous state, or as hydroxyapatite, in nucleus pulposus (NP) that had extruded from intervertebral discs of dogs diagnosed with intervertebral disc extrusion.  It also demonstrates the utility of AFM and SEM for examination of the NP. These findings suggest that calcification plays an important role in disc herniation. The experiments set the stage for a more thorough characterization of the degenerative process and demonstrate how future methods of prevention and treatment could be evaluated. Though informative, the results would be far more compelling if they could be compared to those representing a handful of young, healthy dogs, whether from chondrodystrophic breeds or not.

Specific comments:

Lines 157-158: It would be nice to know the breeds and ages of the dogs from which samples were collected, even if they provide no basis for results correlation.

Lines 184-187: Reference images of all conditions would be helpful, i.e. not containing phosphate, containing amorphous phosphate, and containing significant crystalline hydroxyapatite.

Lines 224-230: Describe the compressive loading in more detail (in this reviewer’s opinion, DMA analysis would be a better way to characterize the properties of NP).  Unconfined compression between smooth platens? What was the approximate sample size (dimensions)? How were they prepared?  How were the dimensions measured?  Were the samples kept hydrated?  Tested at room temperature?

Lines 292-296:  The EDS results should discussed in terms of the expected Ca/P ratio for hydroxyapatite (1.667). 

Figure 7:  Add some interpretation of these spectra, especially in the context of Figure 1.

Lines 311-315: The vertical axis scale of Figure 8 is so large that the stiffness of the samples in the region of interest, up to about 30% strain, cannot be estimated.  It is suggested that Young’s modulus be calculated for each sample (slope of stress-strain up to ~30% strain) and the data presented as mean, range, lower quartile, and upper quartile, or similarly.

Author Response

This study primarily established the presence of calcium phosphate, either in the amorphous state, or as hydroxyapatite, in nucleus pulposus (NP) that had extruded from intervertebral discs of dogs diagnosed with intervertebral disc extrusion.  It also demonstrates the utility of AFM and SEM for examination of the NP. These findings suggest that calcification plays an important role in disc herniation. The experiments set the stage for a more thorough characterization of the degenerative process and demonstrate how future methods of prevention and treatment could be evaluated. Though informative, the results would be far more compelling if they could be compared to those representing a handful of young, healthy dogs, whether from chondrodystrophic breeds or not.

Thank you for your review and your very reasonable assessment of our paper. We agree that it would be interesting to compare our findings to those of young, healthy dogs but that was not the purpose of this study; it is well established that young healthy dogs' nucleus pulposus consists primarily of water giving it a very gelatinous consistency. Our aim with this study is to demonstrate some changes occurring in degenerate discs and in particular some of the techniques which may be helpful in better characterizing these changes. 

Lines 157-158: It would be nice to know the breeds and ages of the dogs from which samples were collected, even if they provide no basis for results correlation.

We agree that this adds an important detail to the results and so we have now reported the breeds and ages of the dogs from whom samples were taken in this study (Lines 245-251).

Lines 184-187: Reference images of all conditions would be helpful, i.e. not containing phosphate, containing amorphous phosphate, and containing significant crystalline hydroxyapatite.

We agree it is helpful to the reader to have such reference images available so we have added a supplementary figure containing smoothed spectra traces of test hydroxyapatite and amorphous calcium phosphate which we feel are useful but would prefer not to incorporate into the main body of the text.

Lines 224-230: Describe the compressive loading in more detail (in this reviewer’s opinion, DMA analysis would be a better way to characterize the properties of NP).  Unconfined compression between smooth platens? What was the approximate sample size (dimensions)? How were they prepared?  How were the dimensions measured?  Were the samples kept hydrated?  Tested at room temperature?

Thank you for your comments and we take on board the possible alternative testing method suggested. We have added some further detail as requested to the methodology of this section (Lines 229-236).

Lines 292-296:  The EDS results should discussed in terms of the expected Ca/P ratio for hydroxyapatite (1.667).

Thank you for this comment, we have now included some discussion of these results (Lines 312-8, 420-2) as well as adding a new citation in support (25).

Figure 7:  Add some interpretation of these spectra, especially in the context of Figure 1.

We have clarified these results and added some interpretation as requested (Lines 322, 331-5).

Lines 311-315: The vertical axis scale of Figure 8 is so large that the stiffness of the samples in the region of interest, up to about 30% strain, cannot be estimated.  It is suggested that Young’s modulus be calculated for each sample (slope of stress-strain up to ~30% strain) and the data presented as mean, range, lower quartile, and upper quartile, or similarly.

We accept that it is difficult to compare stiffness of most of these samples since they appear to behave for the most part very similarly to this testing which we have commented on. Whilst we will in our future work consider calculating Young's modulus as suggested, we feel this is not essential to the presentation of the results in this paper since again we are concerned with demonstrating the potential application of different techniques rather than attempting to draw significant conclusions from any particular set of results. We hope the reviewer understands this explanation in the light of the other revisions we have made which we feel have significantly improved our paper and for which we are very grateful.

Reviewer 2 Report

Comments and Suggestions for Authors

It seems to me to be an interesting, well-structured and innovative work, so it seems to me that it meets all the conditions to be published, as it will be of particular interest to colleagues who work in the area of ​​veterinary neurology.

I only have some minor recommendations:

The title does not seem appropriate to me, given that the work presented is based solely on the analysis of the nucleus pulposus. I suggest that instead of "Applying the techniques of materials science towards an understanding of the process of canine intervertebral disc degeneration." be "Applying the techniques of materials science towards an understanding of the process of canine intervertebral nucleus pulposus degeneration".

Results

Line 235 start by mentioning the number of cases of extruded and non-extruded NP

On Line 236 the acronym HA appears without prior definition

Author Response

The title does not seem appropriate to me, given that the work presented is based solely on the analysis of the nucleus pulposus. I suggest that instead of "Applying the techniques of materials science towards an understanding of the process of canine intervertebral disc degeneration." be "Applying the techniques of materials science towards an understanding of the process of canine intervertebral nucleus pulposus degeneration"

We thank the reviewer for their encouraging review and support and are very pleased they recommend publication. We understand the reservations regarding the title but we would prefer to leave it unchanged since the changes occurring within the nucleus pulposus of the disc are known to be integral to the degenerative process as a whole, and quite possibly hold the key to the pathology of disc extrusion, and we feel that to use 'nucleus pulposus' in the title may limit the potential interest and scope of the likely readership.

Line 235 start by mentioning the number of cases of extruded and non-extruded NP

We have amended this according to the reviewer's suggestion to make the presentation of this section of results clearer.

On Line 236 the acronym HA appears without prior definition

Thank you for noticing this - we have amended.